# Laminarin-Derived from Brown Algae Suppresses the Growth of Ovarian Cancer Cells via Mitochondrial Dysfunction and ER Stress

**DOI:** 10.3390/md18030152

**Published:** 2020-03-09

**Authors:** Hyocheol Bae, Gwonhwa Song, Jin-Young Lee, Taeyeon Hong, Moon-Jeong Chang, Whasun Lim

**Affiliations:** 1Department of Biotechnology, College of Life Sciences and Biotechnology, Korea University, Seoul 02841, Korea; bhc7@korea.ac.kr (H.B.); ghsong@korea.ac.kr (G.S.); 2Department of Pharmacology and Toxicology, Medical College of Wisconsin, Milwaukee, WI 53226, USA; jylee@mcw.edu; 3Department of Food and Nutrition, College of Science and Technology, Kookmin University, Seoul 02707, Korea; taeyeon97@kookmin.ac.kr

**Keywords:** laminarin, ovarian cancer, mitochondria, ER stress, cell death

## Abstract

Ovarian cancer (OC) is difficult to diagnose at an early stage and leads to the high mortality rate reported in the United States. Standard treatment for OC includes maximal cytoreductive surgery followed by platinum-based chemotherapy. However, relapse due to chemoresistance is common in advanced OC patients. Therefore, it is necessary to develop new anticancer drugs to suppress OC progression. Recently, the anticancer effects of laminarin, a beta-1,3-glucan derived from brown algae, have been reported in hepatocellular carcinoma, colon cancer, leukemia, and melanoma. However, its effects in OC are not reported. We confirmed that laminarin decreases cell growth and cell cycle progression of OC cells through the regulation of intracellular signaling. Moreover, laminarin induced cell death through DNA fragmentation, reactive oxygen species generation, induction of apoptotic signals and endoplasmic reticulum (ER) stress, regulation of calcium levels, and alteration of the ER-mitochondria axis. Laminarin was not cytotoxic in a zebrafish model, while in a zebrafish xenograft model, it inhibited OC cell growth. These results suggest that laminarin may be successfully used as a novel OC suppressor.

## 1. Introduction

In the United States, approximately 22,240 human ovarian cancer (OC) cases with 14,070 deaths were reported in 2018. OC is difficult to diagnose and is, therefore, associated with a high mortality rate. The risk of a woman developing OC was 1 in 78 with a diagnosis rate of 11.5 per 100,000 people in the United States between 2010 and 2014 [1]. Epithelial ovarian cancer (EOC), the most frequent (90%) type of OC, is histologically classified as serous, endometrioid, mucinous, clear cell, or undistinguished subtypes [2]. OC is asymptomatic and difficult to diagnose at an early stage [3]. Thus, most OC cases are diagnosed in stage III (51%) and stage IV (29%) with the 5-year survival rates being 42% and 26%, respectively, between 2007 and 2013 [4]. The standard approach for treating EOC is surgery and chemotherapy using paclitaxel and cisplatin [5,6,7]. Yet, EOC relapses in 70–80% cases because of chemotherapy resistance [8]. Therefore, new and safe drugs that are less prone to this problem of chemoresistance are required.

β-Glucan consists of a group of β-d-glucose polysaccharides, which occur in cell walls of bacteria, fungi, yeast, cereals, and seaweeds [9]. The physicochemical activity of β-glucan varies by its source. Basically, β-glucan has a linear backbone with 1–3 β-glycosidic bonds and induces various physiological effects in animals depending on molecular mass, solubility, viscosity, branching structure, and gelation properties [10]. β-Glucan in food has been ingested for a long time and it was first reported in 1981 as having a hypocholesterolemic effect [11]. Additionally, in 1997 the Food and Drug Administration (FDA) released a report that eating at least 3.0 g of β-glucan from oats per day reduced the blood cholesterol levels and heart disease [12]. β-Glucan derived from mushrooms alleviates respiratory tract infection [13]. Dietary intake of β-glucan from brown algae prevents human breast cancer [14]. One of β-glucan, laminarin is a storage glucan existed in brown algae [15]. Laminarin indicates low molecular weight polysaccharide approximately 5 kDa and the basic structure of laminarin consist of (1,3)-β-d-glucopyranose residues with 6-*O* branching part in main and β-(1,6)-interstrand linkages [16]. However, the structure of laminarin is different from species of the source. It possesses diverse biofunctional activities, including anti-inflammatory, anticoagulant, antioxidant, and anticancer properties. Among the anticancer effects, it has been reported effective against colorectal cancer [17,18], melanoma [19], and breast cancer [20]. However, its effects in OC remain unclear.

Therefore, we investigated the effects of laminarin specifically in terms of (i) apoptosis in vitro (ES2 and OV90 cells) and in vivo (zebrafish), (ii) cell cycle progression and reactive oxygen species (ROS) production in vitro, (iii) cytosolic or mitochondrial calcium concentrations and mitochondrial membrane potential (MMP) in vitro, and (iv) intracellular signaling pathways in vitro.

## 2. Results

### 2.1. Laminarin Reduces Cell Proliferation and Induces SubG1 Phase Arrest in EOC Cells

The structure of laminarin consists of poly(β-Glc-(1,3)) with some β-(1,6) interstrand linkages and branch point (Figure 1A). We determined the proliferation of human EOC cells using 5-bromo-2´-deoxyuridine (BrdU) as a DNA synthesis indicator to identify changes induced by laminarin (Figure 1B,C). Laminarin gradually decreased the proliferation of ES2 (by 52.9%; *p* < 0.05) and OV90 (by 63.9%; *p* < 0.001) cells in a dose-dependent manner. Cell cycle assays (Figure 1D,E) revealed an increase in the subG1 population from 5.4% to 20.8% in ES2 cells and from 2.8% to 12.6% in OV90 cells in response to laminarin treatment (0.1, 0.25, 0.5, 1, and 2 mg/mL).

### 2.2. Laminarin Inhibits PI3K/MAPK Intracellular Signaling Pathways in Human EOC Cells

Western blot analyses to study laminarin-induced changes in intracellular signal transduction pathways (PI3K and MAPK) related to cell proliferation (Figure 2) revealed a decrease in the phosphorylation of cyclin D1 (Figure 2A), AKT-P70S6K-S6 (Figure 2B–D), and ERK1/2 (ES2: up to 0.4-fold, *p* < 0.001; OV90: up to 0.3-fold, *p* < 0.01), JNK (ES2: up to 0.2-fold, *p* < 0.01; OV90: up to 0.2-fold, *p* < 0.01), and p38 (ES2: up to 0.2-fold, *p* < 0.001; OV90: up to 0.6-fold, *p* < 0.01) in both OC cell types compared with non-treated cells (Figure 2E–G).

Pretreatment with pathway inhibitors, LY294002 or U0126, almost completely blocked the laminarin-induced decrease in the phosphorylation of AKT and S6 proteins in OC cells (Figure 3A,C). Although p-P70S6K was significantly lower with laminarin in combination with LY294002 or U0126 treatment in ES2 cells, this effect was only seen with U0126 in OV90 cells (Figure 3B). Pretreatment with U0126 or SB203580 also completely blocked the laminarin-induced decrease in ERK1/2 and JNK phosphorylation in both EOC cells (Figure 3D,E). Similarly, pretreatment with SB203580 and LY294002 significantly inhibited the laminarin-induced decrease in p38 protein phosphorylation (Figure 3F).

### 2.3. Laminarin Alters Programmed Cell Death in Human EOC Cells

The terminal deoxynucleotidyl transferase dUTP nick end labeling (TUNEL) assay revealed abundant DNA fragmentation in the nuclei of laminarin-treated ES2 cells and some DNA fragmentation in OV90 cells, but no apoptotic damage in vehicle-treated cells (Figure 4A,B), indicating that laminarin induced programmed cell death. Flow cytometry analysis with annexin V and PI staining of OC cells showed an increase in late apoptotic cells in response to laminarin (Figure 4C,D). ROS assays showed laminarin-induced increase in ROS generation in ES2 and OV90 cells compared with vehicle-treated controls (Figure 4E,F). Western blot data for ES2 and OV90 cells showed a 7.3- and 6.5-fold increase in cleaved caspase-3 and a 1.5- and 2.2-fold increase in caspase-9, respectively (Figure 4G,H). Moreover, laminarin stimulated the release of cytochrome c (ES2: up to 10.6 times, *p* < 0.01; OV90: up to 11.5 times, *p* < 0.01) compared with vehicle-treated control. Collectively, these results suggest that laminarin induces cell apoptosis by increasing DNA fragmentation and apoptosis-related proteins in OC cells.

### 2.4. Effects of Laminarin on Calcium Ion Levels and MMP in EOC Cells

Flow cytometry analysis of human OC cells incubated with fluo-4 or rhod-2 following treatment with laminarin showed a dose-dependent increase in intracellular calcium levels (ES2: 416.5%; *p* < 0.001; Figure 5A and OV90: 449.2%; *p* < 0.001; Figure 5B) and mitochondrial calcium levels (ES2: 372%; *p* < 0.001; Figure 5C and OV90: 286%; *p* < 0.001; Figure 5D) in ES2 and OV90 cells, respectively, compared with control. JC-1 staining indicated that laminarin also induced mitochondrial depolarization (ES2: 3930%; *p* < 0.001; Figure 5E and OV90: 940%; *p* < 0.001; Figure 5F) compared with control.

### 2.5. Laminarin Induces Cell Death and Loss of MMP through Calcium Regulation

Flow cytometry analysis revealed that laminarin (2 mg/mL) increased late apoptotic cells (ES2: 642.9% and OV90: 565.2%), but co-treatment with either of the three calcium chelators—2-APB, BAPTA, and ruthenium red—showed a decrease in late apoptotic cells (ES2: 400.0%, 300.0%, and 400.0%, respectively, in Figure 6A and OV90: 239.1%, 271.4%, and 204.3%, respectively, in Figure 6B). Laminarin increased intracellular calcium levels to 481.8% (*p* < 0.01) in ES2 cells and 217.3% (*p* < 0.05) in OV90 cells compared with the control, but co-incubation with the calcium chelators prevented the laminarin-induced increase in calcium levels (Figure 6C,D). Similarly, mitochondrial calcium levels increased to 785.9% (*p* < 0.001) in ES2 cells and 522.9% (*p* < 0.01) in OV90 cells by laminarin treatment alone, but co-treatment prevented any change in mitochondrial calcium levels (Figure 6E,F). Laminarin induced MMP loss in ES2 cells (up to 392.1%; *p* < 0.01) and OV90 cells (up to 554.5%; *p* < 0.01), but co-incubation with calcium chelators repressed the laminarin-mediated MMP loss (Figure 6G,H).

### 2.6. Laminarin Induces ER Stress in Human EOC Cells

Western blot analysis showed that laminarin stimulated the activation of proteins in the unfolded protein response, including IRE1α (ES2: up to 2.7 times, *p* < 0.01; OV90: up to 3.3 times, *p* < 0.01), ATF6α (ES2: up to 1.8 times, *p* < 0.001; OV90: up to 2.2 times, *p* < 0.01), p-PERK (ES2: up to 2.2 times, *p* < 0.01; OV90: up to 4.5 times, *p* < 0.01), GADD153 (ES2: up to 3.3 times, p < 0.01; OV90: up to 2.1 times, *p* < 0.05), p-eIF2α (ES2: up 4.0 times, *p* < 0.001; OV90: up to 3.4 times, *p* < 0.01), and GRP78 (ES2: up to 2.5 times, *p* < 0.01; OV90: up to 1.8 times, *p* < 0.01) in both OC cell types compared with the control (Figure 7). Taken together, laminarin increased ER stress in ES2 and OV90 cells, leading to cell death.

### 2.7. Laminarin Alters ER-Mitochondrial Mediated Proteins and Autophagy Proteins

Western blot analysis showed that laminarin increased IP3R1 and IP3R2 levels in ES2 and OV90 cells (Figure 8A,B), but decreased GRP75 protein level (Figure 8C). Moreover, laminarin activated MFN2 protein in a dose-dependent fashion in both cell lines (Figure 8D). These results indicated that laminarin stimulated the release of calcium from the ER to the cytosol or mitochondria through ER-mitochondria tethering proteins. In addition, laminarin induced autophagy via inactivation of ULK1 and P62 phosphorylation in ES2 and OV90 cells (Figure 8E,F).

### 2.8. Cytotoxicity of Laminarin in a Zebrafish Xenograft Model

Zebrafish embryos, in which the egg shell and pigment were removed, treated with different doses of laminarin (0.5, 1, and 2 mg/mL) for 24 h did not show any change in viability or development (Figure 9A). When OC cells treated with 1 and 2 mg/mL laminarin were injected into the zebrafish yolk sac in a zebrafish xenograft model, laminarin was found to inhibit tumor formation by 53.3% (*p* < 0.01) and 48.5% (*p* < 0.001) for ES2 cells (Figure 9B), and by 79.0% (*p* < 0.001) and 45.6% (*p* < 0.001) for OV90 cells, respectively (Figure 9C).

## 3. Discussion

Laminarin is a polysaccharide present in brown algae, such as *Laminaria japonica*, *Ecklonia kurome,* and *Eisenia bicyclis* [21]. While its anticancer effects are known against melanoma (SK-MEL-5 and SK-MEL-28) [20,22], hepatocellular carcinoma (HepG2) [23], colon cancer (HT-29, DLD-1, HCT 116 and LoVo) [17,18,19], and prostate cancer (PC-3) [24], its effects against human EOC were demonstrated for the first time.

Laminarin repressed cell proliferation and PI3K/MAPK intracellular signaling pathways and increased the number of late apoptotic cells, subG1 cell population, MMP, apoptotic proteins, ER stress sensor proteins, and ER-mitochondrial tethering proteins in ES2 and OV90 cells. It also suppressed tumor formation in a zebrafish xenograft model.

In the present study, we first confirmed that laminarin repressed OC cell viability. According to previous reports, laminarin suppresses proliferation, colony formation, and migration of human melanoma cells by inhibiting matrix metalloproteinase-2 (MMP-2), MMP-9, and the p-ERK1/2 signaling cascade [22]. Moreover, laminarin represses the proliferation and metastasis of liver cancer cells via a reduction in endogenous hydrogen sulfide production and effects on the pSTAT3/BCL-2 and VEGF cascades, which include MMPs, VEGF, p-AKT, and p-ERK1/2 [23]. Additionally, laminarin induces LoVo cell death by activating pro-apoptotic signaling proteins [17]. Laminarin inhibits the proliferation and increases cell cycle arrest, leading to apoptosis in prostate cancer PC-3 cells by increasing the expression of PTEN and P27kip1 [24]. Consistent with previous results, laminarin induced an accumulation of a subG1 cell population and increased the number of apoptotic OC cells. Laminarin also stimulated the release of cytochrome c from mitochondria and led to caspase-3 and caspase-9 cleavage in ES2 and OV90 cells. These results indicated that laminarin increases the intrinsic apoptosis pathway in OC cells.

Mitochondrial-dependent cell death is associated with mitochondrial calcium concentration overload and mitochondrial membrane permeabilization as well as cleavage of caspases, the release of cytochrome c, and activation of pro-apoptotic proteins [25]. In particular, the mitochondrial calcium overload can affect the release of pro-apoptotic factors by mitochondrial destruction. In addition, mitochondrial dysfunction causes an increase in cytoplasmic calcium concentration in cancer cells, which leads to apoptosis [26]. Our flow cytometry results showed that laminarin stimulated the apoptosis of human OC cells by increasing mitochondrial and cytoplasmic calcium concentrations. Moreover, co-treatment with three calcium chelators-2-APB, BAPTA, and ruthenium red-blocked the effects of laminarin and caused changes in intracellular calcium levels, which led to a decrease in the number of apoptotic ES2 and OV90 cells compared with laminarin treatment alone. Furthermore, regulation of intracellular calcium levels suppressed mitochondrial membrane permeabilization. Laminarin increases intracellular calcium and decreases MMP in human colon cancer cells [18]. To increase the mitochondrial calcium levels, IP3Rs mediate calcium transfer from the ER to the mitochondria [27]. Moreover, the ER-mitochondrial coupling protein GRP75 controls mitochondrial function and calcium homeostasis. Downregulation of GRP75 causes calcium overload in the cytosol and mitochondria, inducing oxidative stress [28]. Another ER-mitochondria tethering protein, MFN2, increases cell death and mitochondrial calcium influx in liver cancer cells [29]. Laminarin activated IP3R1, IP3R2, and MFN2 in ES2 and OV90 cells under the same conditions by increasing mitochondrial calcium level. However, it reduced the expression of GRP75 protein in both OC cell types. Taken together, our results show that laminarin induced cell death by disrupting calcium homeostasis in human OC cells.

Intracellular signal transduction is important in cancer progression, metastasis, and chemotherapy resistance. The PI3K/MAPK pathway serves as a therapeutic target for controlling cancer cell development [30]. Laminarin inhibits the activity of p-AKT and p-ERK1/2 signaling proteins in human liver cancer and melanoma [22,23]. In addition, a blockade of PI3K and MAPK signaling is considered a novel therapeutic strategy to treat OC [31,32]. Likewise, in the present study, laminarin significantly decreased the phosphorylation of PI3K/MAPK signaling proteins in both OC cells. Co-treatment of laminarin with an inhibitor that blocked AKT, ERK1/2, JNK, or P38 eliminated the laminarin-mediated inhibition of PI3K and MAPK signaling in OC cells. These results indicated that laminarin reduced the proliferation of OC cells by inactivating PI3K/MAPK signaling. The ER plays an important role in protein translocation, folding, and post-transcriptional modification in eukaryotic cells [33]. ER stress or unfolded protein response activation decreases cancer cell survival. Moreover, the overexpression of GADD153 by ER stress inhibits PI3K pathway signaling by stimulating the transcription of an AKT inhibitor for decreased cellular proliferation [34,35]. Fucoidan, a component of brown algae, increases the accumulation of IRE1α, ATF6α, and PERK, which are key initiators of ER stress in OC cells [36]. Similarly, in our study, laminarin stimulated ER stress signaling proteins in ES2 and OV90 cells, leading to cell death. Moreover, it regulated autophagy by repressing the phosphorylation of ULK1, which promotes cancer cell survival, proliferation, migration, and invasion [37], and p62, which leads conjugated proteins to degradation [38]. Kaempferol decreased p62 activity in gastric cancer cells and increased LC3-II conversion from LC3-I under cell death conditions [39]. These results indicate that laminarin induces ER stress and autophagosome formation, leading to OC cell death.

## 4. Materials and Methods

### 4.1. Reagents

Laminarin extracted from *Laminaria digitata* (catalog number: L9634) was purchased from Sigma-Aldrich (St. Louis, MO, USA) and dissolved in sterile water before use. The antibodies used are described previously [36].

### 4.2. Cell Culture

ES2 (ovarian clear cell carcinoma cells) and OV90 (papillary serous adenocarcinoma cells) were purchased from American Type Culture Collection (ATCC; Manassas, VA, USA) and incubated as described previously [40].

### 4.3. Cell Proliferation Assay

Proliferation assays were performed using the Cell Proliferation ELISA, BrdU Kit (Cat No. 11647229001, Roche, Basel, Switzerland). Briefly, both cell lines were incubated in a 96-well plate and serum-starved for 24 h in MaCoy’s 5A medium. Cells were then treated with laminarin for 48 h. BrdU was added to the cell culture medium, and the cells were incubated for an additional 2 h at 37 °C. BrdU was detected with an anti-BrdU-POD antibody and a color developed after the addition of the substrate.

### 4.4. TUNEL Assay

Both cell lines were incubated in confocal dishes and starved for 24 h in FBS-free conditions. Cells were incubated with laminarin for 48 h, air-dried, and fixed with 4% paraformaldehyde in phosphate-buffered saline (PBS) for 1 h at room temperature. The cells were briefly rinsed with PBS and permeabilized with 0.1% Triton X-100 in 0.1% sodium citrate for 2 min on ice. Subsequently, the cells were subjected to a TUNEL staining mixture using the In Situ Cell Death Detection kit with tetramethylrhodamine (TMR) red (Roche) for 1 h at 37 °C in the dark. Cells were then washed with PBS and overlaid with DAPI. Fluorescence was detected using an LSM710 (Carl Zeiss, Oberkochen, Germany) confocal microscope.

### 4.5. Annexin V and PI Staining

The induction of apoptosis in OC cells by laminarin was analyzed using the FITC Annexin V Apoptosis Detection Kit I (BD Biosciences, Franklin Lakes, NJ, USA). The cells (5  ×  10^5^ cells) were seeded in 6-well plates and treated with laminarin for 48  h. The supernatants were discarded and the adherent cells were detached with trypsin-ethylenediaminetetraacetic acid (EDTA). The cells were collected by centrifugation, washed with PBS, and resuspended using 1× binding buffer at a dilution of 1  ×  10^6^ cells/mL. Then, 100 μL of the cell suspension (1  ×  10^6^ cells) was transferred to a 5 mL culture tube and incubated with 5  μL FITC Annexin V and 5  μL PI for 15  min at room temperature in the dark. After this, 400 μL of 1× binding buffer was added to the 5 mL culture tube. Fluorescence intensity was determined using a FACS Calibur (BD Biosciences).

### 4.6. Cell Cycle Assay

The cells were seeded in 6-well plates and treated with various doses in laminarin for 48 h at 37 °C in a CO_2_ incubator. The supernatants were discarded, and the adherent cells were detached with trypsin-EDTA. The cells were collected by centrifugation, washed with PBS, and resuspended using 1 × binding buffer at a dilution of 1  ×  10^6^ cells/mL. Then, 100 μL of the cell suspension (1  ×  10^6^ cells) was transferred to a 5 mL culture tube and incubated with 5  μL of RNase A and 5  μL of PI for 30  min at room temperature in the dark. After this, 300 μL of 1× binding buffer was added to the 5 mL tube. The intensity of the fluorescence was determined using a FACS Calibur flow cytometer (BD Biosciences).

### 4.7. Cellular ROS Determination

Intracellular ROS production was estimated using 2′,7′-dichlorofluorescein diacetate (DCFH-DA, Sigma), which is converted to fluorescent 2′,7′-dichlorofluorescein (DCF) in the presence of peroxides. Cells were detached with trypsin-EDTA, collected by centrifugation, and washed with PBS. The cells were treated with 10 μM DCFH-DA for 30 min at 37 °C, washed twice with PBS, and treated with different doses of laminarin for 1 h at 37 °C in a CO_2_ incubator. The treated cells were then washed with PBS again. Fluorescent DCF intensity was analyzed using a flow cytometer (BD Biosciences).

### 4.8. Measurement of Intracellular Concentrations of Free Calcium

ES2 and OV90 cells were seeded in 6-well plates and incubated for 24 h in serum-free medium when the cells reached 70% to 80% confluency. The cells were then treated with different concentrations of laminarin for 48 h at 37 °C in a CO_2_ incubator. Supernatants were discarded, and adherent cells were detached with trypsin-EDTA. The cells were collected by centrifugation, resuspended in 3 μM fluo-4 acetoxymethyl ester (Invitrogen, Carlsbad, CA, USA), and incubated at 37 °C in a CO_2_ incubator for 20 min. The stained cells were washed with PBS, and the fluorescence intensity was analyzed using a flow cytometer (BD Biosciences).

### 4.9. Measurement of Mitochondrial Calcium Concentration

Both cell lines were seeded in 6-well plates and incubated for 24 h in serum-free medium when cells reached 70–80% confluency. The cells were then treated with different concentrations of laminarin for 48 h at 37 °C in a CO_2_ incubator. Supernatants were discarded, and adherent cells were detached with trypsin-EDTA. The cells were collected by centrifugation, resuspended in 3 μM rhod-2, and incubated at 37 °C in a CO_2_ incubator for 20  min. The stained cells were washed with PBS and the fluorescence intensity was analyzed using a flow cytometer (BD Biosciences).

### 4.10. JC-1 MMP Assay

Changes in the JC-1 MMP were determined using a Mitochondria Staining Kit (Cat no: CS0390, Sigma). The cells were seeded in 6-well plates and incubated for 24 h in serum-free medium until they reached 70% confluency. Then, cells were treated with laminarin in a dose-dependent manner for 48 h at 37 °C in a CO_2_ incubator. Supernatants were discarded, and adherent cells were detached with trypsin-EDTA. The cells were collected by centrifugation, resuspended in a staining solution, which included 200 × JC-1 and 1 × staining buffer, and incubated at 37 °C in a CO_2_ incubator for 20 min. The stained cells were collected by centrifugation and washed once with 1 × JC-1 staining buffer. The cells were then centrifuged once more and resuspended in 1 mL staining buffer. Fluorescence intensity was analyzed using a FACSCalibur (BD Biosciences).

### 4.11. Western Blot Analysis

The protein concentrations in whole-cell extracts were determined using the Bradford Protein Assay (Bio-Rad, Hercules, CA, USA) with BSA as the standard. Proteins were denatured, separated by sodium dodecyl sulfate-polyacrylamide gel electrophoresis (SDS-PAGE), and transferred to nitrocellulose membranes. Blots were developed using enhanced chemiluminescence detection and quantified by measuring the intensity of light emitted from correctly sized bands under ultraviolet light using a ChemiDoc EQ system and Quantity One software (Bio-Rad). Immunoreactive proteins were detected using goat anti-rabbit polyclonal antibodies against phosphorylated and total proteins at a 1:1000 dilution and separated by 10% SDS-PAGE. As a loading control, total protein and α-tubulin (TUBA) were used to normalize results for the detection of target proteins. Multiple exposures of each western blot were used to ensure the linearity of chemiluminescent signals.

### 4.12. Toxicity Assays Using a Zebrafish Model

To confirm laminarin toxicity, embryos were exposed to 0.003% phenylthiourea (PTU) for 14 h to suppress pigmentation, the embryo’s shell was removed, and the embryo was treated with various concentrations of laminarin in 24-well plates. After 24 h, embryo viability and development were observed under light microscopy (Carl Zeiss Stereo microscope DV4). Pictures were taken by fixing zebrafish embryos onto a glass slide with 3% methylcellulose (Sigma-Aldrich).

### 4.13. Xenografts

Xenograft studies were performed according to previous experiments (Mazumder et al., 2018) with modifications. Fertilized eggs were treated with Danieau’s solution containing 0.003% PTU at 28.5 °C for 48 h to suppress pigmentation. Micropipettes were created from a 1.0 mm glass capillary (World Precision Instruments, FL, USA) by using a micropipette puller (Shutter Instrument, Novato, CA, USA) for injection and anesthesia. At 48 h after fertilization, zebrafish were anesthetized in 0.02% tricaine (Sigma) and immobilized on an agar plate. Cells were treated with laminarin in a dose-dependent manner for 22 h and stained for an additional 2 h with 4 μM of CellTracker CM-Dil Dye (Invitrogen). Cells (100-200) were injected into the yolk sac by microinjection (PV820 microinjector, World Precision Instruments). Subsequently, zebrafish were incubated in 24-well plates containing Danieau’s solution with 0.003% PTU at 28.5 °C for 72 h. Fish were immobilized in a drop of 3% methylcellulose in Danieau’s solution on a glass slide. Pictures were taken using fluorescence microscopy (Leica DE/DM 5000B, Wetzlar, Germany). Fluorescent tumors were quantified by ImageJ software (National Institutes of Health, Bethesda, MD, USA).

### 4.14. Statistical Analyses

All quantitative data were subjected to a least-squares analysis of variance using the General Linear Models procedure in the Statistical Analysis System program (SAS Institute Inc. Cary, NC, USA). Western blot data were corrected for differences in sample loading using total protein or TUBA data as a covariate. All tests of significance were performed using the appropriate error terms according to the expectation of the mean squares for the error. A *p*-value less than 0.05 was considered significant. Data are presented as least-square means with standard errors.

## 5. Conclusions

In summary, our results provide the first evidence that laminarin has anticancer effects in human OC cells. Laminarin increased cytosolic and mitochondrial calcium levels in human OC cells through the regulation of ER-mitochondrial proteins. Laminarin also decreased PI3K and MAPK signaling in OC cells, whereas it activated ER stress, leading to OC cell death. Zebrafish xenograft model results showed that laminarin prevents tumor formation within the embryo yolks. Our results suggest that laminarin may serve as a novel therapeutic for the treatment of OC.

## Figures and Tables

**Figure 1 marinedrugs-18-00152-f001:**
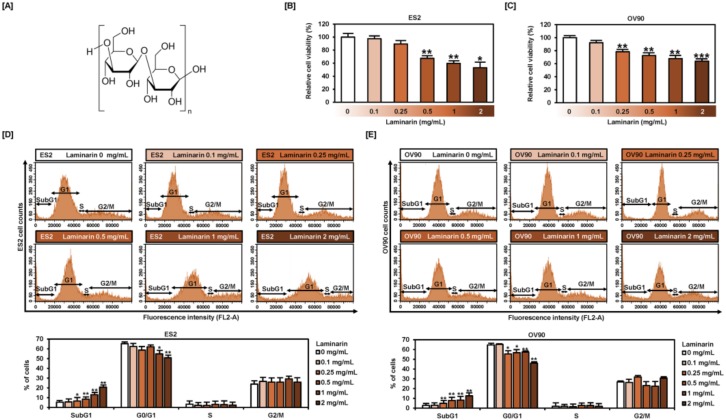
Cell viability and cell cycle progression in laminarin-treated ES2 and OV90 cells. (**A**) Structure of laminarin derived from *Laminaria digitate*. (**B**,**C**) Cell proliferation analysis performed using BrdU reveals that laminarin (0.1, 0.25, 0.5, 1 and 2 mg/mL) decreased cell proliferation in a dose-dependent manner in ES2 (**B**) and OV90 (**C**) cells. Results are calculated compared with vehicle-treated control (viability considered as 100%). (**D**,**E**) Alterations in cell cycle progression were analyzed by flow cytometry using propidium iodide (PI) staining in laminarin-treated ES2 (**D**) and OV90 (**E**) cells. The percentages in each cell cycle stage are shown. *** *p*  <  0.001, ** *p*  <  0.01, * *p*  <  0.05 compared with vehicle-treated control cells.

**Figure 2 marinedrugs-18-00152-f002:**
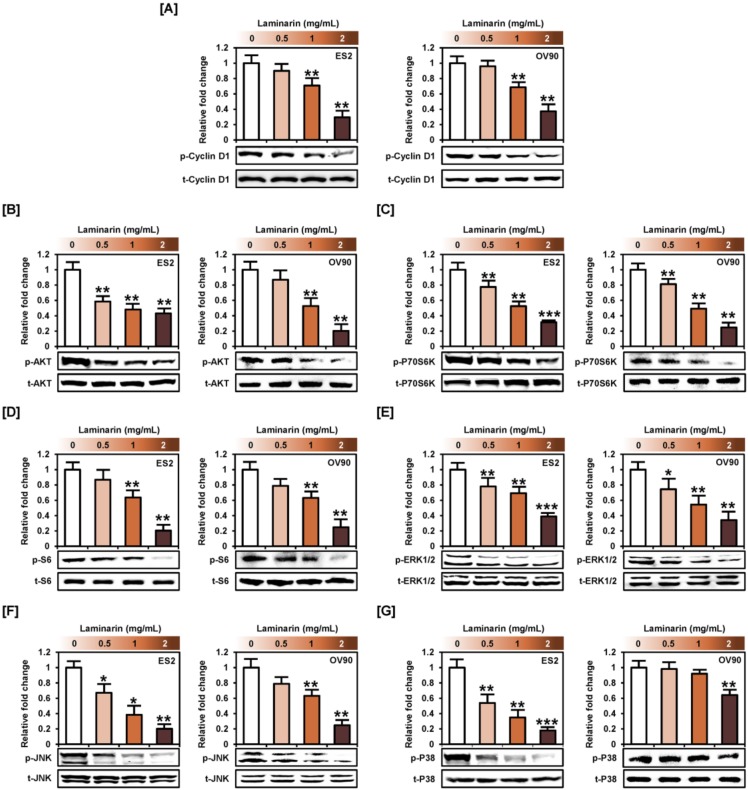
Laminarin inhibited intracellular signal transduction in ovarian cancer (OC) cells. (**A**–**G**) Immunoblotting showing the phosphorylation of cyclin D1 (**A**), AKT (**B**), P70S6K (**C**), S6 (**D**), extracellular signal-regulated kinase 1/2 (ERK1/2) (**E**), c-Jun N-terminal kinase (JNK) (**F**), and P38 (**G**) proteins in laminarin (0.5, 1, and 2 mg/mL)-treated OC cells. Phosphoprotein intensities were normalized to the total protein levels compared with vehicle-treated controls. *** *p*  <  0.001, ** *p*  <  0.01, and * *p*  <  0.05 indicate statistical significance compared with non-treated cells.

**Figure 3 marinedrugs-18-00152-f003:**
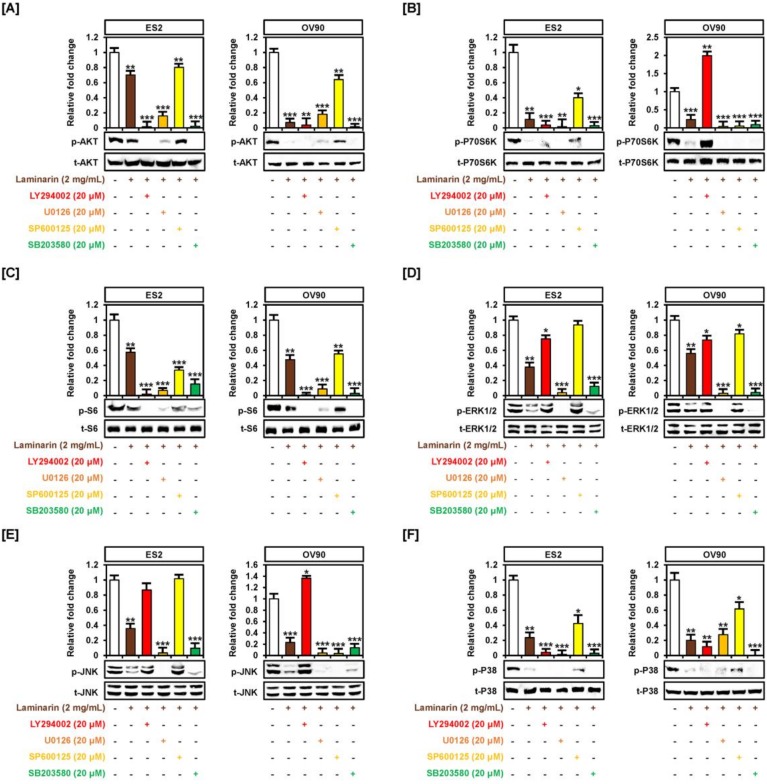
Inhibition of PI3K and MAPK signals in laminarin-treated OC cells. OC cells were treated with inhibitors, including LY294002 (20  μΜ, AKT), U0126 (10  µM, ERK1/2), SB203580 (20  µM, p38), SP600125 (20  µM, JNK), and laminarin (2 mg/mL) for western blot analysis to determine levels of AKT (**A**), P70S6K (**B**), S6 (**C**), ERK1/2 (**D**), JNK (**E**), and P38 (**F**). Phosphoprotein intensities were normalized to the total protein levels compared with vehicle-treated controls. *** *p*  <  0.001, ** *p*  <  0.01, and * *p*  <  0.05 indicate statistical significance compared with non-treated cells.

**Figure 4 marinedrugs-18-00152-f004:**
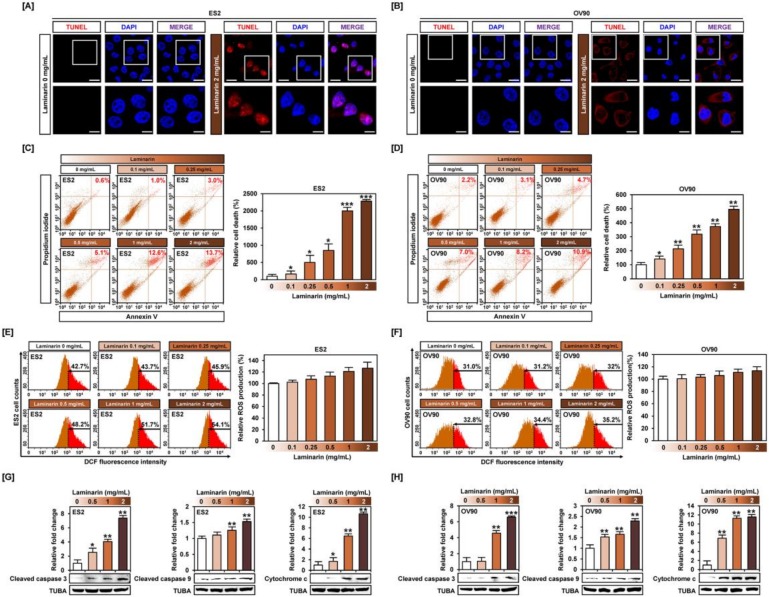
Laminarin induced apoptosis of human OC cells. (**A**,**B**) DNA fragmentation was observed using terminal deoxynucleotidyl transferase dUTP nick end labeling (TUNEL) staining (red). The nuclei of cells were counterstained using 4′,6-diamidino-2-phenylindole (DAPI) (blue). The scale bar represents 20 μm (in the first horizontal panel set) and 5 μm (in the second horizontal panel set). The apoptotic ES2 (**C**) and OV90 (**D**) cells treated with laminarin were measured using annexin V and propidium iodide (PI) fluorescent dyes. Reactive oxygen species (ROS) production in laminarin-treated ES2 (**E**) and OV90 (**F**) cells was observed using dichlorofluorescein (DCF) fluorescence by flow cytometry compared with vehicle-treated cells. (**G**,**H**) Apoptotic signals induced by laminarin were observed using western blot analysis in OC cells. Apoptotic protein intensities were normalized to alpha-tubulin (TUBA) compared with vehicle-treated cells. *** *p*  <  0.001, ** *p*  <  0.01, and * *p*  <  0.05 indicate statistical significance compared with non-treated cells.

**Figure 5 marinedrugs-18-00152-f005:**
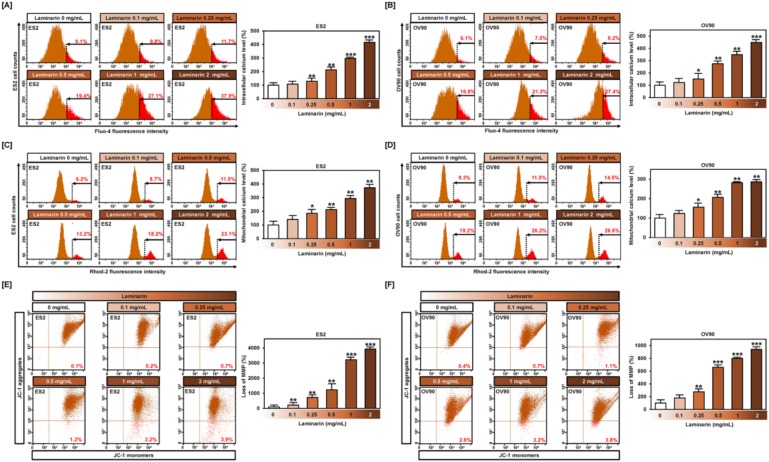
Effects of laminarin on calcium homeostasis and mitochondria membrane potential (MMP) in ES2 and OV90 cells. (**A**,**B**) Flow cytometry analysis using fluo-4 showed cytosolic calcium in response to laminarin (0.1, 0.25, 0.5, 1, and 2 mg/mL). Flow cytometry analysis using rhod-2 revealed mitochondrial calcium levels in ES2 (**C**) and OV90 (**D**) cells following laminarin treatment for 48 h. Flow cytometry analysis indicated MMP in ES2 (**E**) and OV90 (**F**) cells that were treated with various laminarin concentrations. Loss of MMP was analyzed using JC-1 red and green fluorescence ratios. *** *p*  <  0.001, ** *p*  <  0.01, and * *p*  <  0.05 indicate statistical significance compared with non-treated cells.

**Figure 6 marinedrugs-18-00152-f006:**
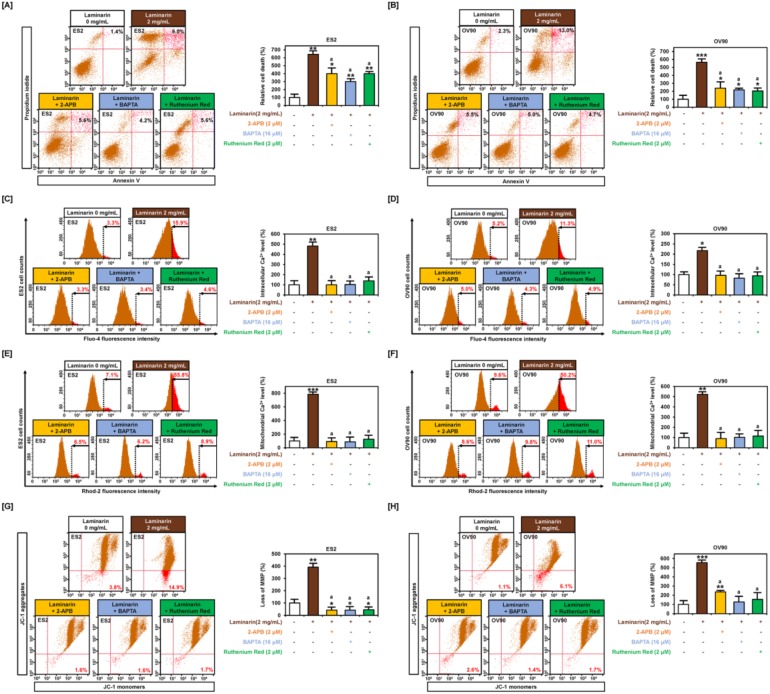
Inhibition of calcium influx in laminarin-treated OC cells. The number of late apoptotic ES2 (**A**) and OV90 (**B**) cells co-treated with laminarin and calcium chelators (2-aminoethoxydiphenyl borate (2-APB), 1,2-*bis*(2-aminophenoxy)ethane-*N*,*N*,*N*′,*N*′-tetraacetic acid tetrakis(acetoxymethyl ester) (BAPTA), and ruthenium red) was measured using annexin V and PI staining. (**C**,**D**) Flow cytometry analysis using fluo-4 showed cytosolic calcium in OC cells co-treated with laminarin and calcium chelators. (**E**,**F**) Flow cytometry analysis using rhod-2 revealed mitochondrial calcium levels in ES2 and OV90 cells following treatment with laminarin and calcium chelators. (**G**,**H**) Loss of MMP was analyzed using JC-1 red and green fluorescence ratios by treatment of ES2 and OV90 cells with laminarin and a calcium chelator. *** *p*  <  0.001, ** *p*  <  0.01, and * *p*  <  0.05 indicate statistical significance compared with non-treated cells. ‘a’ indicates statistical significance (*p* < 0.05) compared with laminarin-treated cells.

**Figure 7 marinedrugs-18-00152-f007:**
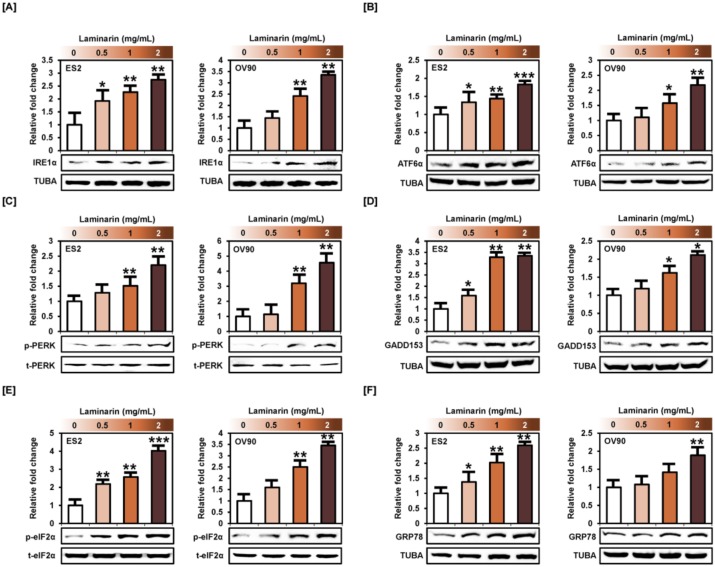
Laminarin induced ER stress in OC cells. (**A**–**F**) Activation of ER stress-associated proteins, including inositol-requiring enzyme-1α (IRE1α), activating transcription factor 6α (ATF6α), growth arrest and DNA damage 153 (GADD153), protein kinase R (PKR)-like endoplasmic reticulum kinase (PERK), eukaryotic translation initiation factor 2α (eIF2α), and 78-kDa glucose-regulated protein (GRP78), was analyzed by western blotting of ES2 and OV90 cells treated with various concentrations of laminarin (0.5, 1 and 2 mg/mL). Target protein intensity was detected and analyzed relative to total protein or TUBA protein. *** *p*  <  0.001, ** *p*  <  0.01, and * *p*  <  0.05 indicate statistical significance compared with non-treated cells.

**Figure 8 marinedrugs-18-00152-f008:**
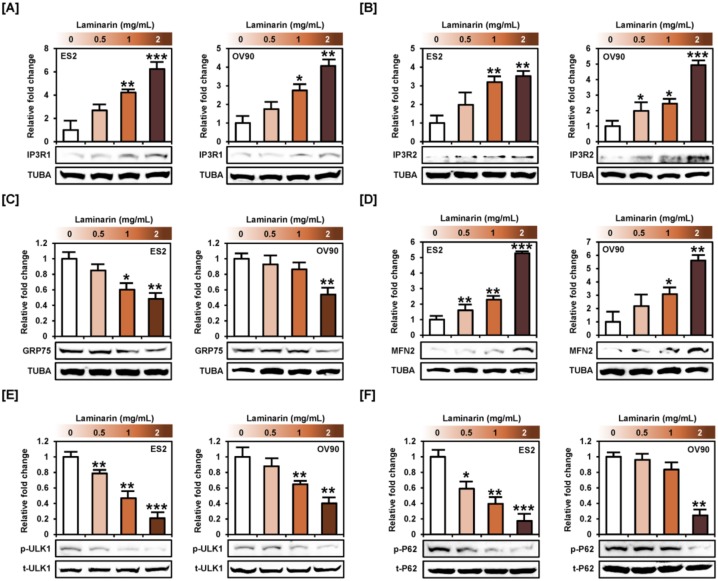
Laminarin altered intracellular signaling related to ER-mitochondria contact proteins and autophagy. (**A**–**D**) Expression of ER-mitochondrial tethering proteins, including IP3R1, IP3R2, GRP75, and MFN2, was analyzed by western blot analysis of ES2 and OV90 cells treated with various concentrations of laminarin (0.5, 1, and 2 mg/mL). (**E**,**F**) Expression of autophagic proteins, including ULK1 and p62, was analyzed by western blot analysis of ES2 and OV90 cells. Target protein intensity was detected and analyzed relative to total protein or TUBA protein. *** *p*  <  0.001, ** *p*  <  0.01, and * *p*  <  0.05 indicate statistical significance compared with non-treated cells.

**Figure 9 marinedrugs-18-00152-f009:**
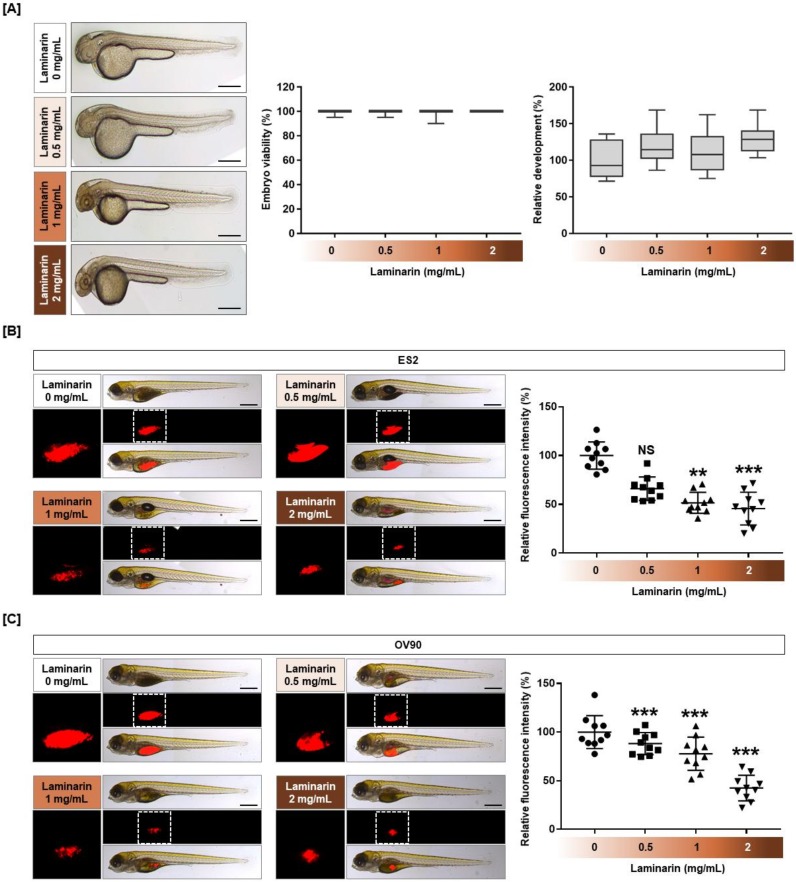
Effects of laminarin on cytotoxicity and tumor formation in vivo. (**A**) Zebrafish embryos, with their egg shells removed, were treated with various concentrations of laminarin (0.5, 1, and 2 mg/mL) for 24 h. Under light microscopy, normal zebrafish viability and development were observed following laminarin treatment. (**B**,**C**) Laminarin-treated OC cells were injected into zebrafish yolks to form a xenograft model. Zebrafish were incubated in 24-well plates containing Danieau’s solution with 0.003% phenylthiourea at 28.5 °C for 72 h. CM-Dil dye stained the tumor cells, and they were quantified by ImageJ software. *** *p*  <  0.001, ** *p*  <  0.01, and * *p*  <  0.05 indicate statistical significance compared with non-treated embryos.

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
