# Peer review of "Laminarin-Derived from Brown Algae Suppresses the Growth of Ovarian Cancer Cells via Mitochondrial Dysfunction and ER Stress"

_marinedrugs, 2020, doi:10.3390/md18030152_

Round 1

Reviewer 1 Report

The manuscript is devoted to the study of anticancer activity of brown algal polysaccharide laminarin. Authors described that this non-toxic polysaccharide can be used as effective suppressor of ovarian cancer. However, there is a lack of characterization of laminarin as chemical compound, i.e. its purity, composition and structure. Moreover, molecular mass is also important for polysaccharides. There is a high variability in structural properties of natural polysaccharides including those from algae. Authors should add more information about this laminarin based on information from producer, own investigations or from research reports elsewhere.

I also recommend to add structural formula of laminarin as Figure 1. In addition to this, the discussion part should be expanded by comparing of laminarin activities with those of other beta-glucans, for example from fungal sources. Why authors use laminarin in testing instead of well known glucans from yeast or mushrooms? Have laminarin some benefits as marine drug in economical or medicinal points of view?

Authors should check the quality of colour figures 4-6 and improve the resolution if need.

Author Response

Reviewer 1

The manuscript is devoted to the study of anticancer activity of brown algal polysaccharide laminarin. Authors described that this non-toxic polysaccharide can be used as effective suppressor of ovarian cancer. However, there is a lack of characterization of laminarin as chemical compound, i.e. its purity, composition and structure. Moreover, molecular mass is also important for polysaccharides. There is a high variability in structural properties of natural polysaccharides including those from algae. Authors should add more information about this laminarin based on information from producer, own investigations or from research reports elsewhere.

Response: We appreciate the reviewer’s valuable comments and suggestions on our manuscript. We have substantially revised our manuscript according to the reviewer’s suggestions. To address the reviewer’s comments, we prepared a point-by-point response to each comment of the reviewer and highlighted changes in text of the manuscript in yellow. We added the information for laminarin including composition and structure in Introduction and Materials and Methods section and inserted the laminarin structure in Figure 1A.  

I also recommend to add structural formula of laminarin as Figure 1. In addition to this, the discussion part should be expanded by comparing of laminarin activities with those of other beta-glucans, for example from fungal sources. Why authors use laminarin in testing instead of well known glucans from yeast or mushrooms? Have laminarin some benefits as marine drug in economical or medicinal points of view?

Response: According to reviewer’s comment, we added laminarin structure in Figure 1A. One of our research projects is to investigate therapeutic effects of component of seaweed on progression of ovarian cancer. Therefore, we used laminarin derived from Laminaria digitate. The production of seaweed including larminaria such as Saccharina japonica and Undaria pinnatifida in South Korea ranks first and second among domestic seaweeds, and exports are on the rise worldwide. Based on productivity of larminarin easily, it is predicted that the economic income will increase after validation of therapeutic effects of laminarin for development of agent.

Authors should check the quality of colour figures 4-6 and improve the resolution if need.

Response: We checked the resolution of Figure 4-6. The figures were made as following guideline of Marine Drugs.

Reviewer 2 Report

Overall very interesting and important paper, needs serious improvement in Introduction, as no relevant info about glucan was offered. The methods, results and discussion sections are fine and description of novel effects of glucan is important and interesting.

However, Instroduction needs serious improvements - the major topic of the paper is glucan, but this section offers almost no information on what glucan is, what was done during 50 years of intensive research, why was glucan even studies and why exactly this one.  Laminarin is little bit neglected glucan with sometimes questionable results, so it must be discussed. In addition, line 47-48 is not true, laminarin has been tested repeatedly and interesting results were published e.g., here: Int J Biol Macromol. 2007 Mar 10;40(4):291-8

Author Response

Reviewer 2

Overall very interesting and important paper, needs serious improvement in Introduction, as no relevant info about glucan was offered. The methods, results and discussion sections are fine and description of novel effects of glucan is important and interesting.

However, Introduction needs serious improvements - the major topic of the paper is glucan, but this section offers almost no information on what glucan is, what was done during 50 years of intensive research, why was glucan even studies and why exactly this one.  Laminarin is little bit neglected glucan with sometimes questionable results, so it must be discussed. In addition, line 47-48 is not true, laminarin has been tested repeatedly and interesting results were published e.g., here: Int J Biol Macromol. 2007 Mar 10;40(4):291-8

Response: We appreciate the reviewer’s valuable comments and suggestions on our manuscript. We substantially revised Introduction section including the explanation for glucans. Moreover, we revised the indicated sentence.

Round 2

Reviewer 1 Report

The revised manuscript is now acceptable for publication in Marine Drugs.

Reviewer 2 Report

The revised version significantly improved the quality.